# Effect of Daily Intake of Green Tea Catechins on Cognitive Function in Middle-Aged and Older Subjects: A Randomized, Placebo-Controlled Study

**DOI:** 10.3390/molecules25184265

**Published:** 2020-09-17

**Authors:** Yoshitake Baba, Shun Inagaki, Sae Nakagawa, Toshiyuki Kaneko, Makoto Kobayashi, Takanobu Takihara

**Affiliations:** 1Central Research Institute, ITO EN, Ltd., 21 Mekami, Makinohara, Shizuoka 421-0516, Japan; shu-inagaki@itoen.co.jp (S.I.); sae-nakagawa@itoen.co.jp (S.N.); m-kobayasi@itoen.co.jp (M.K.); t-takihara@itoen.co.jp (T.T.); 2Tokyo Skytree Station Medical Clinic, Ryobi Building F4 33-13 Mukojima 3-chome, Sumida-ku, Tokyo 131-0033, Japan; t.kaneko.19790402@gmail.com

**Keywords:** green tea, catechins, cognitive function, Cognitrax, middle-aged, randomized placebo-controlled trial

## Abstract

Epidemiological studies in Japan, including the Nakajima study and the Tsurugaya study, have indicated that green tea consumption may improve cognitive impairment. Catechins, which are typical polyphenols contained in green tea, have been reported to have antioxidative, anti-inflammatory, and neuroprotective effects. However, their impact on human cognitive function remains unclear. Therefore, we performed a double-blind, randomized, controlled study to investigate the effect of 336.4 mg of decaffeinated green tea catechins (GTC) on cognitive function after a single dose and after 12 weeks of daily intake. This study included Japanese adults between the ages of 50 and 69 years with a Mini-Mental State Examination Japanese version score of >24 and self-assessed cognitive decline. The Cognitrax testing battery was used to evaluate cognitive function. The incorrect response rate on the Continuous Performance Test significantly decreased after a single dose of GTC. After 12 weeks of daily GTC intake, the response time for Part 4 of the 4-part Continuous Performance Test, which is a two-back test, was shortened. These results suggest that daily intake of GTC might have beneficial effects on working memory.

## 1. Introduction

Dementia is a growing worldwide health problem among the elderly [1]. In 2015, Japan’s aging rate was 26.6%, the highest among developed countries [2]. According to the Hisayama study [3,4], the crude prevalence of Alzheimer’s disease in Japan increased tenfold from 1985 to 2012. Considering this context, age-related declines in cognitive function and increases in the prevalence of dementia have become important public health issues in Japan.

Previous epidemiological studies have shown that consumption of green tea may improve impairments in cognitive function [5,6,7,8]. According to a study in Nakajima-machi, Ishikawa Prefecture, elderly people with normal cognitive function who frequently drank green tea for an average of 4.9 years had a lower odds ratio (0.32) of developing dementia and mild cognitive impairment (MCI) than those who did not drink green tea [5]. Kuriyama et al. reported that people who drank two or more cups of green tea per day had a lower prevalence of cognitive impairment than people who drank less than three cups per week, and the cognitive impairment could not fully be explained by risk factors for cerebrovascular dementia [6].

In our previous study in elderly patients, we found that scores on the revised Hasegawa’s dementia scale (HDS-R) [7] and the Mini-Mental Status Examination Japanese Version (MMSE-J) [8] were improved after powdered green tea was ingested daily for 12 months. Of note, the MMSE-J and HDS-R are both interactive cognitive function tests in which an evaluator asks a question in words and the test subject answers in words. Both are used as simple tests to evaluate cognitive function.

In another study, we used the more complex Cognitrax test battery to investigate the effects of matcha on cognitive function [9]. Matcha is high-quality tea; before being harvested, the tea leaves are covered and cultivated for about 20 days, leading to an increase in its amino acid content. Cognitrax is a personal computer (PC)-based evaluation consisting of 10 individual tests that assess memory, attention, facial expression recognition, working memory, visual information processing, and motor function. After reading the explanation for each test on a PC, subjects answer by operating a keyboard, with one practice session prior to each test. An advantage of Cognitrax is that it can also be used to measure reaction time. In addition, the pattern of each administered test question using Cognitrax is random, while the MMSE-J assesses the same questions in the same order during each administration. In our previous study, we used Cognitrax to demonstrate that matcha intake significantly reduced the simple response time in the Stroop Test (ST) and the correct response time in the Shifting Attention Test (SAT). There was also a significant increase in the number of correct responses in the SAT [9].

Previous studies have suggested that powdered green tea and matcha have positive effects on attention function [9]; however, persistent questions remain. For example, we still do not know whether catechins, the main component of green tea, solely contribute to improvements in cognitive function. There have, however, been many in vivo studies related to the effect of catechins on Alzheimer’s disease and on age-related cognitive decline [10,11,12,13]. These studies have reported that (−)-epigallocatechin gallate (EGCG), a major catechin in green tea, suppresses Aβ production and deposition, which is thought to be a major factor related to the cognitive dysfunction seen in Alzheimer’s disease [10,11,14]. Studies using senescence-accelerated mouse models have also shown that EGCG inhibits declines in cognitive function [10] and learning ability [11,12]. In addition, EGCG has been shown to inhibit social withdrawal and declines in spatial reasoning ability [13], both associated with age-related cognitive dysfunction. There have, however, been no reports demonstrating whether catechins alone can improve attention function. Instead, it has been shown that when theanine and caffeine are co-administered, theanine interacts with caffeine to enhance its effect on simple reaction time, information-processing speed, mental fatigue, and general fatigue [15]. Of note, in our previous matcha-related study, the administered substance contained caffeine, theanine, and catechin; therefore, the sole effect of catechins on human cognitive function remains unclear.

Another persistent question relates to the intake period required for green tea and matcha-related improvements in cognitive function. Our previous study [9] showed a positive effect on cognitive function after 12 weeks of continuous matcha ingestion, which is consistent with a previous epidemiological study showing that a higher frequency of green tea intake led to less cognitive impairment. However, the acute effect of catechin intake on cognitive function remains unclear.

Therefore, the main purpose of this study was to use Cognitrax to clarify whether catechins alone contribute to the improvements in attention that we observed in our previous study. Furthermore, we aimed to investigate whether catechins affected other cognitive functions besides attention and whether a single dose of catechins versus 12 weeks of daily ingestion resulted in different effects on cognitive function.

## 2. Results

The study flow diagram is shown in Figure 1. After consideration of study inclusion and exclusion criteria, 52 patients were enrolled in the study. After allocation, one person in the placebo group and one person in the catechin group violated the exclusion criteria. Three people in the catechin group dropped out of the study during the test period due to personal reasons. The final analysis set was comprised of 25 subjects (12 males and 13 females) in the placebo group and 22 subjects (12 males and 10 females) in the catechin group. The clinical characteristics of the patients for this study are shown in Table 1. For ethical reasons, the intake of beverages containing polyphenols was not limited. The study participants also drank tea in their daily lives. Before the intervention, we investigated the amount of green tea consumed by the participants in their daily lives and found that the placebo group consumed 5.24 ± 5.6 cups/week and the matcha group consumed 8.43 ± 9.6 cups/week. No significant difference was observed between the two groups. Therefore, the participants took the test sample and also consumed approximately one cup of green tea per day.

### 2.1. MMSE-J (Interactive Test)

The average MMSE-J (backwards task) score before the intervention (baseline) was 27.6 ± 1.8 in the placebo group and 27.8 ± 1.3 in the catechin group. Post-intervention, the average score was 28.0 ± 2.0 in the placebo group and 28.0 ± 1.7 in the catechin group. No significant differences were observed between the two groups before or after the intervention.

### 2.2. Cognitrax Test (PC-Based Cognitive Function Test)

The Cognitrax cognitive tests were administered in the following order: Verbal Memory Test (VBM), Visual Memory Test (VIM), Finger-tapping Test (FTT), Symbol Digit-coding Test (SDC), ST, SAT, Continuous Performance Test (CPT), Perception of Emotions Test (POET), Non-verbal Reasoning Test (NVRT), and 4-part Continuous Performance Test (FPCPT). The VBM and VIM were then repeated, with the results of the first VBM and VIM tests considered to evaluate immediate memory and the results of the second VBM and VIM tests considered to evaluate delayed memory. There was an interval of about 50 min between the first VBM/VIM and the second VBM/VIM. Performance on memory tasks was evaluated using results from the VBM and VIM (Table 2). Performance on attention tasks was evaluated using results from the ST, SAT, CPT, and FPCPT Parts 1 and 2 (Table 3). Performance on facial expression recognition tasks was evaluated using results from the POET (Table 4). Performance on working memory tasks was evaluated using results from the FPCPT Parts 3 and 4 (Table 5). Performance on visual information-processing tasks was evaluated using results from the SDC and NVRT (Table 6). Performance on motor function tasks was evaluated using results from the FTT (Table 7).

#### 2.2.1. Performance on Memory Tasks

Green tea catechins (GTC) administration, whether after a single dose or after chronic ingestion, had no significant effects on VBM or VIM results.

#### 2.2.2. Performance on Attention Tasks

The catechin group had significantly lower commission errors on the CPT than the placebo group after a single dose. There were no significant differences in other attention-related tests.

#### 2.2.3. Performance on Facial Expression Recognition Tasks

We evaluated the perception of positive emotions (calm, happiness) and negative emotions (sadness, anger) using the POET, which evaluates a sum of positive and negative emotions, as well as positive and negative emotions independently. Using the POET, there were no significant differences in the facial expression recognition task between the placebo group and the catechin group, either after a single dose or following chronic ingestion of GTC.

#### 2.2.4. Performance on Working Memory Tasks

After repeated doses of GTC, the mean correct response time on the FPCPT Part 4, a two-back test, was significantly lower in the catechin group than in the placebo group. Using the FPCPT Part 3, a one-back test, there were no significant differences between the placebo group and the catechin group. After 12 weeks of daily ingestion of GTC, the reaction time on one working memory task was improved.

#### 2.2.5. Performance on Visual Information Processing Tasks

Using the SDC and NVRT, there were no significant differences between the placebo group and the catechin group. Catechins had no effect on visual information processing tasks either after a single dose or following chronic ingestion.

#### 2.2.6. Performance on Motor Function Tasks

Using the FTT, there were no significant differences in the mean number of hits with the right or the left hands between the placebo group and the catechin group. Catechins had no effect on motor function tasks either after a single dose or following chronic ingestion.

### 2.3. Blood Biomarkers

There were no significant differences in the serum levels of amyloid-β1-40 or amyloid-β1-42 or the Aβ1-40/Aβ1-42 ratio between the placebo group and the catechin group (Table 8). There were no significant differences in the levels of the secreted form of amyloid-β precursor protein α (sAPPα) or the amyloid-β precursor protein (APP)770 between the placebo group and the catechin group. There were also no significant differences in the serum levels of brain-derived neurotrophic factor (BDNF) between the placebo group and the catechin group. Figure 2 shows the serum BDNF levels in the two groups at baseline and after the 12-week study period.

## 3. Discussion

In our previous study, we found that administration of 2 g of matcha over 12 weeks significantly reduced the simple reaction time in the ST and the correct response time in the SAT and significantly increased the number of correct responses in the SAT [9]. The matcha used in that study contained a combination of caffeine, theanine, and catechins, and our results were likely related to interactions between these components. In this study, we specifically focused on catechins, the largest component of green tea, and examined whether catechin intake influenced attention function. We further investigated whether catechin intake affected cognitive functions other than attention. In this study, a single ingestion showed improvement in attention function, and long-term ingestion also had a positive effect on working memory.

Previous studies revealed that catechins improve cognitive function [7,8,9]; the catechin dose used in the studies contacted by Kataoka et al. [7], Ide et al. [8], and the authors’ [9] was 174 mg, 227 mg, and 171 mg, respectively. It is presumed that cognitive function was improved by the combined effects of catechin, theanine, and caffeine in the green tea. In order to clarify the effect of catechins alone on cognitive function, the dose of catechins used in this study was 336.4 mg. That is twice the amount of the catechin dose used in previous studies.

The subjects in this study had MMSE-J scores of 24 points or higher. Since the maximum MMSE-J cutoff value distinguishing healthy people from people with MCI is reported to be 27 or 28, and the maximum MMSE-J cutoff value distinguishing people with MCI from people with Alzheimer’s disease is reported to be 23 or 24 [16], this study may have included some patients with MCI, though this diagnosis was not evaluated by a doctor. However, MCI is said to be recoverable: the revert rate is high at 14 to 44% [17]. In a pilot study by Ide et al. [8], a significant improvement in the MMSE score was observed when a subject with an MMSE score of 15.3 ± 7.7 ingested green tea powder for 12 weeks. Hence, catechins may contribute to this increase in reversion.

When healthy subjects take 100 mg of EGCG orally, the peak plasma concentration of free EGCG is reached after 1 to 2 h [18,19]. In this study, the Cognitrax test was started about 50 min after ingesting the test food, and the Cognitrax test continued for approximately 50 min. Therefore, it took approximately 100 min from ingestion of test food to completion of the Cognitrax test. Hence, it was considered that a single dose of GTC had an effect after absorption. However, no effect was observed. Further investigation is needed to understand the difference between the effects of single and continuous administration.

This study assessed the effects of single and repeated doses of GTC. The CPT commission errors decreased significantly after a single dose of GTC. According to previous studies, the consumption of flavan-3-ols increased energy metabolism [20] and blood adrenaline concentrations [21], which probably stimulated the sympathetic nervous system via sensory nerves in the digestive tract. Caffeine is known to transiently improve performance and has been shown to reduce reaction times for attention tasks [22]. In this study, the amount of caffeine ingested in a single dose was only 2.7 mg, an amount that was unlikely to produce an effect. Thus, it was surmised that caffeine had no effect on test results. Given that blood adrenaline levels have been shown to be significantly elevated even 2 h after administration of Flavan-3-ols [23], the reduction in the number of incorrect responses after a single dose of GTC is assumed to be due to the potent action of catechins on the sympathetic nervous system, though the underlying mechanism still remains unknown.

Single doses of GTC had no effect on working memory-related tasks. In a related study, Borgwardt et al. administered milk whey-based beverages mixed with 0.05% green tea extract (0.023–0.026% polyphenols, 0.0025–0.005% caffeine, 0.00015–0.0006% theobromine, and 0.0005–0.0015% theanine) to healthy people aged 21 to 28 years and instructed them to perform an n-back task. Although they found no significant differences in incorrect answer rates or reaction times, they did report activation of parts of the brain devoted to working memory, including the dorsolateral prefrontal cortex (R), inferior parietal lobule (L, R), and middle frontal gyrus (L) on functional magnetic resonance imaging [23]. Given that the subjects consumed green tea extract, which contained multiple components including catechins, the results were assumed to have been mainly due to the effects of theanine and caffeine. When caffeine and theanine are administered in combination, caffeine has been reported to behave differently due to its interaction with theanine [15]. In agreement with Borgwardt’s study, we found no differences between the number of incorrect answers and the reaction times after a single dose. Scholey et al. conducted electroencephalogram studies after administration of 300 mg EGCG and reported that α, β, and γ waves were activated 120 min after administration [24]. In this study, Cognitrax was conducted between 50 and 100 min after ingestion of catechin, a little earlier than what was reported in Scholey et al. Based on a study by Borgwardt et al., increase of brain activity was observed after single ingestion. Conversely, we assumed that there would be reduction in average correct response time on the 2-back task after repeated consumption of GTC. This study noted improved performance in the 2-back task only after repeated doses, which suggests that intake of GTC over a certain duration is necessary to improve performance. After absorption, polyphenols are rapidly conjugated, and those that are not absorbed are metabolized by intestinal bacteria [25]. It has been suggested that the metabolites of (+)-catechin and (−)-epicatechin may directly act on the brain because they traverse the blood brain barrier (BBB) [25]. Metabolites of EGCG have also been reported to pass through the BBB and have antioxidant activity and neuroitogenic activity [26]. Therefore, it is assumed that daily intake is an action mediated by the gut-Brain axis. However, further investigation is needed to determine how brain function is changed with daily intake.

The antioxidant activities of catechins are important to consider when considering their effects on cognitive function. Catechins have radical-scavenging ability [27], so they are thought to reduce oxidative stress. In vivo studies have shown that catechins inhibit accumulation of oxidative damage to DNA in the brain and suppress cognitive decline [28,29]. Ide et al. [30] reported a significant decline in serum levels of malondialdehyde-modified low density lipoprotein (MDA-LDL) after 12 weeks of daily intake of 2 g of powdered green tea. The subjects’ average MMSE-J score was 15.7 in the placebo group and 15.9 in the green tea group, indicating moderate dementia. While no improvements in MMSE-J scores were observed in that study, the lowered MDA-LDL levels suggested that daily intake of catechins reduced oxidative stress and protected brains that were vulnerable to oxidative stress. The oxidative state of the brain and its relationship to cognitive function are also topics for future research studies.

This study measured serum BDNF, Aβ(1–40), Aβ(1–42), APP770, and sAPPα levels as biomarkers related to cognitive function disorders. By activating potassium channels, sAPPα may hyperpolarize nerve cells and thereby protect them [31]. Accordingly, this level was measured as an indicator of normal APP metabolism. Furthermore, a study has shown that secretion of APP770 may increase due to vascular endothelium inflammation [32] and may reflect the state of nerve cells and blood vessels; therefore, APP770 levels were measured as a risk predictor for vascular dementia. Since plasma Aβ concentrations have been associated with progression of dementia [33], these levels were measured to predict the risk of dementia in healthy middle-aged and elderly people whose cognitive function was starting to decline. As shown in Table 8, the plasma Aβ (1–40) and Aβ (1–42) levels and the Aβ (1–42)/Aβ(1–40) ratio were measured after 12 weeks of GTC intake, with no significant differences identified between the placebo group and the catechin group. We also did not identify any significant differences in the levels of plasma APP770 or sAPPα between the catechin group and the placebo group.

BDNF is an indicator of a healthy brain environment and may have important functions in memory formation and learning [34]. BDNF is abundant in the central nervous system but can also be found in the blood; its concentrations in the cerebrospinal fluid might correlate with serum concentrations [35]. Furthermore, patients with Alzheimer’s disease have been reported to have a positive correlation between BDNF concentrations and their MMSE-J score [36]. In an in vivo study, when a forced swimming test was performed after restraint stress, the swimming time was shown to be reduced, with suppressed reductions in BDNF levels [37]. Conversely, in a chronic cerebral hypoperfusion rat model, administration of EGCG reduced swimming times in the water maze test, increased superoxide dismutase activity, and decreased MDA levels, though it did not increase BDNF levels [38]. The catechin group showed a significantly higher value than that in the baseline (Wilcoxon signed-rank test, *p* = 0.007), though there was no significant difference between the control group and the catechin group after 12 weeks of daily intake. Figure 2 shows the change in serum BDNF between baseline and daily intake for 12 weeks. A study using SH-SY5Y cells reported that the addition of catechins increased both the length and the number of neurons [39]. However, in this study, there were no significant differences between the control group and the catechin group after 12 weeks of daily intake. Therefore, the effects of catechin intake on BDNF levels and neuron numbers require further investigation.

This study demonstrated the differences in the effects of daily intake of catechins and the observed changes in blood BDNF, Aβ (1–40), Aβ (1–42), APP770, and sAPPα levels in healthy middle-aged and elderly subjects. No effects of continuous catechin intake or changes from baseline were observed. Further research is needed to diagnose cognitive function using these markers. However, in future studies, it will be important to identify background levels of these markers in healthy subjects without cognitive impairment.

The small sample size and inclusion of only Japanese individuals in the study population were the limitations of this study. Thus, further studies with various ethnicities are needed to generalize the applicability of the findings.

In this study, intake of 336.4 mg of GTC beneficial effects on working memory in subjects, a finding that differed from our previous study in which subjects ingested 2 g of matcha. In future studies, we would like to verify the interactions between the different components of green tea, including theanine, catechin, and caffeine, and to further investigate how green tea improves cognitive function. Furthermore, we would like to clarify how daily catechin intake affects cognitive function. While this study examined self-assessed cognitive decline subjects between 50 and 69 years of age, the effect of catechin intake on patients with reduced cognitive function also needs to be investigated.

## 4. Materials and Methods

### 4.1. Ethical Considerations

The study protocols were examined and approved by the Ethics Committees of Nihonbashi Egawa Clinic (Tokyo, Japan; approval number: food-18071705). This study was conducted in accordance with the tenets of the Declaration of Helsinki (adopted in 1964, amended in Fortaleza in October 2013), the Ethical Guidelines for Medical and Health Research Involving Human Subjects (Ministry of Education, Culture, Sports, Science and Technology and the Ministry of Health, Labour and Welfare Notice No. 3 of 2014), and the Act on the Protection of Personal Information (May 30, 2003, Law No. 57). The study was conducted at the Tokyo Skytree Station Medical Clinic (Tokyo, Japan) from August 2018 to December 2018. The study was registered at the University hospital Medical Information Network (UMIN, Tokyo, Japan) as UMIN000033813. At the first visit, the principal investigator fully explained the details of this study directly to the subjects and obtained voluntary written consent from subjects to participate in the study.

### 4.2. Subjects

Subjects were recruited among healthy Japanese men and women aged 50 to 69 years with self-assessed declines in cognitive function. The MMSE-J was performed on 60 potentially eligible subjects. Patients were considered eligible for inclusion if they met the following criteria: (1) could reliably take three capsules daily for 12 continuous weeks, (2) had a MMSE-J score of 24 or more, and (3) were non-smokers. Patients were excluded if they met any of the following criteria: (1) were currently taking any medications or undergoing any outpatient treatments, (2) had a history of or complications from serious liver, kidney, endocrine, cardiovascular, gastrointestinal, lung, blood, or metabolic diseases, (3) had a history of drug and/or food allergies, (4) were using health foods and/or supplements that could affect cognitive function, (5) were taking medications that could affect cognitive function, (6) had extremely unbalanced diets or extremely irregular lifestyle habits related to diet or sleep, (7) had suspected insomnia, (8) had a current or previous history of psychiatric disorders or alcoholism, (9) were currently participating in another clinical trial or had participated within the prior 3 months, (10) had an irregular employment schedule, such working night shifts, or (11) were judged to be inappropriate for the study as determined by the principal investigator.

Enrolled subjects were randomly assigned to the placebo or catechin groups using a random-numbers table, with age, gender, and MMSE-J score used as allocation factors. Subjects performed both the serial sevens task and the backwards task in the MMSE-J, and allocation was based on the score for the backwards task. The randomization process was performed at HUMA R&D CORP. (Contract Research Organization) in Japan.

### 4.3. Study Design

The study was implemented with a double-blind, placebo-controlled, parallel-group design. The primary endpoints included MMSE-J and Cognitrax results. The secondary endpoints included serum levels of Aβ (1–40), Aβ (1–42), sAPPα, APP770, and BDNF.

Subjects took three placebo or three catechin capsules per day for 12 weeks and were instructed to take the capsules after breakfast. If they did not eat breakfast, they were instructed to take them in the morning. The intake record was carried out by a web input system at HUMA R&D CORP., using a PC. The subjects were not required to restrict their intake of other polyphenols (green tea, black tea, oolong tea, etc.) and or to deviate from their usual diets. During the test period, subjects were prohibited from taking any other health foods, supplements, or medications that could affect cognitive function. In addition, they were instructed to avoid excessive exercise or dietary restrictions, binge eating, or heavy drinking. Although there were no specific restrictions on other types of health foods or supplements, subjects were encouraged to avoid intake of these products as much as possible. If they did take any of these products, the types and amounts were documented on a recording sheet. If they were taking any medications, they were also required to record the names and dosages of the drugs. Other than these limitations, the subjects were told to maintain their normal lifestyle. Subjects were also asked to fill out a daily diary entry to document their intake status and the presence of any adverse events, respiratory infections, abdominal pain, etc.

### 4.4. Test Food

THEA-FLAN 90S (ITO EN Ltd., Tokyo, Japan), consisting of catechins isolated from green tea extract, was used as the test food. THEA-FLAN 90S contains more than 90% polyphenols. During daily intake of THEA-FLAN 90S (482 mg), the catechin content was 336.4 mg and the caffeine content was 2.7 mg. The breakdown of catechins in THEA-FLAN 90S is EGCG 216.9 mg, (−)-epicatechin gallate 96.4 mg, (−)-epigallocatechin 3.2 mg, (−)-epicatechin 1.7 mg, (−)-gallocatechin gallate 12.5 mg, (−)-catechin gallate 4.0 mg, (−)-gallocatechin 1.0 mg, and (−)-catechin 0.8 mg. Number 1 porcine gelatin brown capsules were filled with THEA-FLAN 90S for use in this study. The placebo used the same capsules filled with brown-colored corn starch. Both catechin and placebo capsules used corn starch as the excipient. The capsules were manufactured by Sunsho Pharmaceutical Co., Ltd. (Shizuoka, Japan).

### 4.5. Patient Evaluations

The MMSE-J and blood biomarkers were measured at baseline and at 12 weeks after ingestion. Body weight, heart rate, systolic and diastolic blood pressures, and the Cognitrax test battery were measured at baseline, after a single dose, and after 12 weeks of daily ingestion. Hematologic tests (white blood cell count, red blood cell count, hemoglobin, hematocrit, platelet count, mean corpuscular volume, mean corpuscular hemoglobin, mean corpuscular hemoglobin concentration, total protein, triglycerides, total cholesterol, high-density lipoprotein cholesterol, low-density lipoprotein cholesterol, alkaline phosphatase, aspartate aminotransferase, alanine aminotransferase, γ-guanosine triphosphate, lactate dehydrogenase, uric acid, urea nitrogen, total bilirubin, albumin, creatinine, blood sugar, and hemoglobin A1c measurements) were obtained at baseline and after 12 weeks of ingestion for safety evaluations at SRL, Inc. (Tokyo, Japan).

During the single-dose evaluation, subjects arrived at the hospital and consumed the test meal. They then underwent interviews, body measurements, and the Cognitrax test, with the Cognitrax test starting about 50 min after ingestion. For the 12-week multi-dose evaluation, the test food was not ingested on the same day as the test battery. After continuous intake for 12 weeks, a questionnaire, physical measurements, and the Cognitrax test were performed. The Cognitrax test started at about 50 min, similar to the single-dose test. The following prohibitions were enforced before visiting the hospital for tests: (1) patients were required to refrain from prolonged exercise or intense exercise causing shortness of breath starting from the day prior to the test, (2) patients were required to avoid overeating, excessive dieting, sleep deprivation, excessive exertion deviating from normal daily activities, and drinking alcohol starting from the day prior to the test, and (3) patients were instructed to consume no food or drinks (with the exception of water) starting 6 h prior to the test because the baseline was measured and blood samples were collected upon arrival.

#### 4.5.1. MMSE-J

The MMSE-J (Nihon Bunka Kagakusha Co., Ltd. Tokyo, Japan) is an interactive cognitive function test that was used for this study. This assessment is a Japanese version of the MMSE [40], and its validity and reliability have been established on a Japanese population [41]. This test assesses the following cognitive tasks: time orientation, location orientation, memorization, attention and calculation, recall, naming, repetition, three-stage commands, reading, writing, and copying, and it is evaluated by a total score. Two types of tasks were performed during the attention and calculation section, including the serial sevens task and the backwards task, with the score for the backwards task used for allocation.

#### 4.5.2. Cognitrax Test

Cognitrax is a PC-based cognitive function test developed by CNS Vital Signs (Morrisville, NC, USA) and provided to the Japanese market by Health Solution, Inc. (Tokyo, Japan) [42]. It is comprised of the following 10 test items: (1) VBM, in which patients memorize 15 words and are then asked to select the words that they remember from 30 words that appear randomly; (2) VIM, a graphic version of the VMT; (3) FTT, in which patients quickly tap a key for 10 s with both their right and left index fingers; (4) SDC, in which patients enter a number corresponding to a symbol with reference to a symbol-to-number correspondence table; (5) ST, which consists of Parts 1 through 3. In Part 1, patients press a key when a character appears (red, yellow, blue, and green letters written in black are displayed.) Parts 2 and 3 display the letters red, yellow, blue, and green written in red, yellow, blue, and green. In Part 2, patients press a key when the meaning of letters and colors match. In Part 3, patients press a key when letters and colors do not match. (6) SAT, in which patients follow instructions on a screen and select the option that matches the color or shape, including combinations of red, blue, yellow, and green letters and colors; (7) CPT, in which letters from the alphabet are randomly displayed one-by-one. Subjects are asked to press a key only when the letter B is displayed; (8) POET, in which a photo of a face and a written description of a facial expression are displayed. The facial expression is described using four words, including “calm,” “happy,” “sad,” and “anger.” Subjects press a key if the photo and description match; (9) NVRT, in which one of the four sections is left blank and three have symbols. Five examples are displayed on the screen. The subjects look at the examples and choose the ones that are closest to the three symbols; and (10) FPCPT, which consists of Part 1 through Part 4. For Part 1, subjects press a key when a figure appears. For Part 2, subjects press a key when a green circle appears. Part 3 is one-back task, and Part 4 is a two-back task. Shapes are displayed for Parts 3 and 4. Shapes are a combination of circular, triangular, square, and star figures and red, blue, yellow, and green colors.

#### 4.5.3. Blood Biomarkers

On the day of the tests, subjects were prohibited from eating from 6 h before arrival at the hospital until completion of the tests. At the visit, height and weight were measured, and blood samples were then collected. Blood for BDNF assessments was collected in serum blood collection tubes, and blood for assessment of Aβ (1–40), Aβ (1–42), sAPPα, and APP770 levels was collected in EDTA-2Na blood collection tubes. After collection, the blood was centrifuged at 3000 rpm and dispensed into 1.5-mL Eppendorf tubes. Measurement were performed using a kit diluted as follows: Aβ(1–40) was diluted 20 times with the Human Amyloid-β(1–40) (FL) Assay kit (IBL, Gunma, Japan), Aβ (1–42) was diluted four times with the Human Amyloid-β(1–42) (FL)Assay kit (IBL), sAPPα was diluted four times with the sAPPα (highly sensitive) Assay kit (IBL), APP770 was diluted 50 times with the Human APP770 Assay kit (IBL), and BDNF was diluted 20 times with the BDNF, Human, ELISA Kit Quantikine (R&D Systems, Minneapolis, MN, USA). Any measured values below the kit range were excluded due to inaccuracy. Table 8 shows the number of samples for each measurement item. The measurements were carried out by Skylight Biotech Inc. (Akita, Japan).

### 4.6. Statistical Analysis

Values are reported as mean ± SD. Data normality was assessed using the Shapiro-Wilk test. If normality was confirmed, unpaired *t*-tests were used to evaluate comparisons, and if nonnormality was confirmed, Mann-Whitney *U* tests were used to evaluate comparisons. Data were analyzed with SAS 9.4 (SAS Institute Inc., Cary, NC, USA). Considering the multiplicity of tests, the significance level for inter-condition differences was adjusted to *p* < 0.05/3 = 0.017 using a Bonferroni correction.

## Figures and Tables

**Figure 1 molecules-25-04265-f001:**
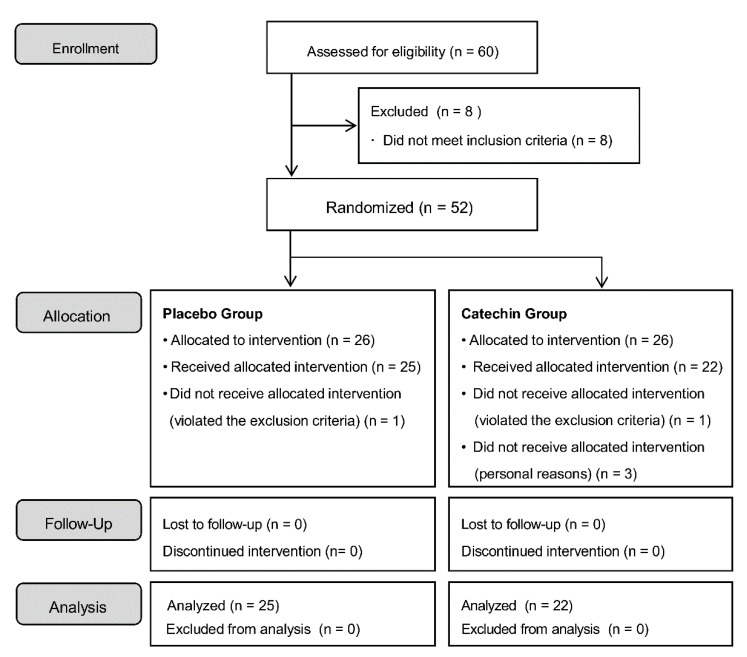
Study flow diagram.

**Figure 2 molecules-25-04265-f002:**
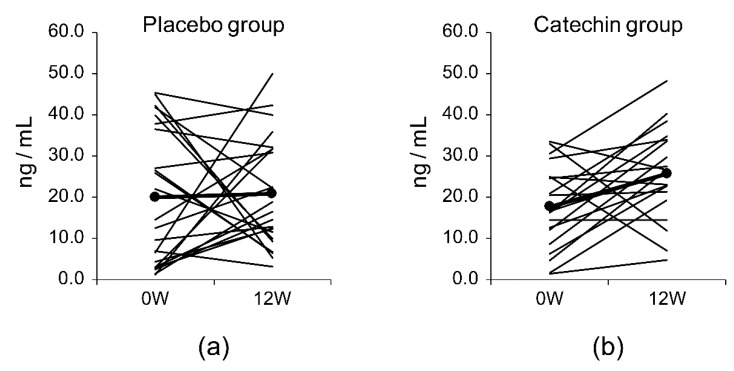
Serum BDNF levels over 12 weeks (W) in the (**a**) placebo group and (**b**) catechin group.

**Table 1 molecules-25-04265-t001:** Clinical characteristics of the patients.

Characteristic	Placebo	Catechin	*p* Values
Number of subjects	26	26	
Sex (M/F)	13/13	13/13	1.00
Age	58.1 ± 6.0	58.1 ± 6.5	1.00
Hight (cm)	161.3 ± 7.2	163.0 ± 8.0	0.49
Weight (kg)	55.8 ± 8.1	58.3 ± 7.5	0.33
BMI	21.4 ± 2.5	21.8 ± 1.5	0.25

Values are presented as mean ± SD. M, male; F, female; BMI, body mass index.

**Table 2 molecules-25-04265-t002:** Effects of green tea catechins on memory-related tasks.

Task			−1 Week (Baseline)	*p* Values	0 Week (Single Dose)	*p* Values	12 Weeks	*p* Values
VBM	Correct hits immediate	Placebo	11.2 ± 1.7		11.5 ± 1.4		11.2 ± 2.2	
		Catechin	11.3 ± 1.9	0.484	11.4 ± 1.7	0.839	11.6 ± 1.1	1.000
	Correct passes immediate	Placebo	14.8 ± 0.5		14.8 ± 0.4		14.6 ± 0.7	
		Catechin	14.5 ± 1.6	0.533	14.7 ± 0.4	0.630	14.5 ± 1.7	0.428
	Correct hits delay	Placebo	10.1 ± 2.4		10.0 ± 2.2		10.0 ± 3.0	
		Catechin	10.8 ± 2.3	0.272	10.8 ± 2.0	0.158	10.4 ± 2.2	0.604
	Correct passes delay	Placebo	14.5 ± 0.7		14.0 ± 1.4		14.3 ± 1.1	
		Catechin	14.0 ± 1.2	0.162	14.1 ± 1.4	0.647	13.7 ± 1.8	0.242
VIM	Correct hits immediate	Placebo	10.4 ± 2.1		10.1 ± 1.8		10.0 ± 2.0	
		Catechin	10.7 ± 1.8	0.690	10.5 ± 1.9	0.488	10.7 ± 1.6	0.239
	Correct passes immediate	Placebo	12.5 ± 2.0		12.4 ± 2.0		12.2 ± 2.2	
		Catechin	12.2 ± 1.9	0.623	12.7 ± 1.6	0.673	12.5 ± 1.3	0.914
	Correct hits delay	Placebo	10.4 ± 2.1		10.4 ± 2.0		9.0 ± 2.5	
		Catechin	10.3 ± 2.4	0.903	10.4 ± 1.9	0.932	10.3 ± 1.5	0.046
	Correct passes delay	Placebo	11.4 ± 2.4		11.4 ± 2.6		10.8 ± 2.5	
		Catechin	11.5 ± 2.0	0.955	11.0 ± 2.1	0.239	11.0 ± 2.0	0.957

VBM, Verbal Memory Test; VIM, Visual Memory Test. Values are presented as mean ± SD. Correct hits and correct passes are per capita figures.

**Table 3 molecules-25-04265-t003:** Effects of green tea catechins on attention-related tasks.

Task			−1 Week (Baseline)	*p* Values	0 Week (Single Dose)	*p* Values	12 Weeks	*p* Values
ST (Part 1)	Simple reaction time (ms) [A]	Placebo	355 ± 74		374 ± 174		341 ± 52	
		Catechin	357 ± 73	0.660	348 ± 78	0.877	360 ± 79	0.522
ST (Part 2)	Complex reaction time correct (ms) [B]	Placebo	684 ± 107		670 ± 108		676 ± 81	
		Catechin	671 ± 85	0.942	655 ± 81	0.796	661 ± 74	0.587
ST (Part 3)	Stroop reaction time correct [C]	Placebo	779 ± 111		749 ± 117		766 ± 117	
		Catechin	770 ± 115	0.615	737 ± 91	0.889	748 ± 80	0.790
	Stroop commission errors	Placebo	1.23 ± 1.4		1.88 ± 3.7		1.24 ± 1.2	
		Catechin	0.77 ± 0.99	0.228	0.74 ± 0.96	0.072	0.77 ± 0.97	0.175
	(C/B) × 100	Placebo	115 ± 12		112 ± 8.2		114 ± 12	
		Catechin	115 ± 11	0.906	113 ± 8.6	0.738	114 ± 9.0	0.982
	(A/B) × 100	Placebo	52.2 ± 8.3		56.1 ± 23		50.9 ± 7.7	
		Catechin	53.1 ± 7.1	0.656	52.8 ± 7.3	0.499	54.4 ± 9.3	0.166
SAT	Correct responses	Placebo	43.2 ± 7.4		45.4 ± 9.5		48.4 ± 5.8	
		Catechin	48.5 ± 5.4 *	0.005	50.6 ± 5.5	0.027	50.5 ± 7.2	0.274
	Errors	Placebo	6.1 ± 4.1		5.0 ± 5.1		3.7 ± 3.0	
		Catechin	3.3 ± 2.5 *	0.007	2.9 ± 1.9	0.213	3.4 ± 2.9	0.812
	Correct reaction time (ms)	Placebo	1183 ± 162		1141 ± 172		1094 ± 134	
		Catechin	1118 ± 110	0.097	1064 ± 125	0.086	1059 ± 146	0.401
CPT	Correct responses	Placebo	39.8 ± 0.6		39.7 ± 0.6		38.8 ± 3.4	
		Catechin	40.0 ± 0.0	0.161	39.8 ± 0.4	0.738	39.9 ± 0.3	0.249
	Omission errors	Placebo	0.15 ± 0.61		0.28 ± 0.61		1.16 ± 3.4	
		Catechin	0.00 ± 0.0	0.161	0.17 ± 0.39	0.738	0.09 ± 0.29	0.249
	Commission errors	Placebo	0.77 ± 1.6		0.68 ± 1.2		0.44 ± 0.87	
		Catechin	0.19 ± 0.40	0.218	0.04 ± 0.21 *	0.004	0.18 ± 0.39	0.483
	Choice reaction time correct (ms)	Placebo	488 ± 49		499 ± 54		498 ± 51	
		Catechin	484 ± 59	0.615	481 ± 61	0.085	489 ± 59	0.359
FPCPT (Part 1)	Average correct response time (ms)	Placebo	397 ± 110		397 ± 125		371 ± 50	
		Catechin	385 ± 96	0.756	389 ± 115	0.765	422 ± 134	0.670
FPCPT (Part 2)	Correct responses	Placebo	6.0 ± 0.2		6.0 ± 0.0		6.0 ± 0.2	
		Catechin	6.0 ± 0.0	0.336	6.0 ± 0.0	1.000	6.0 ± 0.0	0.371
	Average correct response time (ms)	Placebo	451 ± 56		445 ± 47		450 ± 46	
		Catechin	443 ± 63	0.459	450 ± 67	0.893	452 ± 53	1.000
	Incorrect responses	Placebo	0.19 ± 0.57		0.12 ± 0.33		0.32 ± 1.2	
		Catechin	0.08 ± 0.27	0.604	0.09 ± 0.29	0.726	0.14 ± 0.47	0.765
	Average incorrect response time (ms)	Placebo	56.0 ± 165		47.5 ± 132		51.4 ± 143	
		Catechin	46.4 ± 173	0.681	41.1 ± 136	0.815	43.0 ± 141	0.796
	Omission errors	Placebo	0.04 ± 0.20		0.00 ± 0.00		0.04 ± 0.20	
		Catechin	0.00 ± 0.00	0.336	0.00 ± 0.00	1.000	0.00 ± 0.00	0.371

ST, Stroop Test; SAT, Shifting Attention Test; CPT, Continuous Performance Test; FPCPT, 4-part Continuous Performance Test. Values are presented as mean ± SD. *P* values are calculated using unpaired *t*-tests or Mann–Whitney *U* tests and Bonferroni correction; * *p* < 0.05/3 = 0.017 vs. placebo group.

**Table 4 molecules-25-04265-t004:** Effects of green tea catechins on facial expression recognition-related task.

Task			−1 Week (Baseline)	*p* Values	0 Week (Single Dose)	*p* Values	12 Weeks	*p* Values
POET	Correct responses	Placebo	10.6 ± 1.6		11.3 ± 0.8		10.4 ± 1.7	
		Catechin	10.5 ± 1.3	0.564	11.0 ± 1.4	0.402	10.7 ± 1.0	0.904
	Average correct reaction time (ms)	Placebo	1190 ± 187		1176 ± 173		1163 ± 169	
		Catechin	1131 ± 195	0.276	1113 ± 160	0.198	1151 ± 195	0.827
	Omission errors	Placebo	1.4 ± 1.6		0.7 ± 0.8		1.6 ± 1.7	
		Catechin	1.5 ± 1.3	0.564	1.0 ± 1.4	0.402	1.3 ± 1.0	0.904
	Commission errors	Placebo	3.9 ± 2.1		3.4 ± 2.5		3.0 ± 1.9	
		Catechin	4.7 ± 2.9	0.275	4.0 ± 2.8	0.472	3.3 ± 2.5	0.622
Positive Emotions	Correct hits	Placebo	5.5 ± 0.9		5.8 ± 0.4		5.4 ± 1.0	
		Catechin	5.2 ± 0.9	0.134	5.7 ± 0.7	0.838	5.5 ± 0.7	0.865
	Reaction time	Placebo	1175 ± 212		1167 ± 198		1128 ± 188	
		Catechin	1125 ± 186	0.493	1116 ± 139	0.757	1158 ± 178	0.502
Negative Emotions	Correct hits	Placebo	5.1 ± 0.8		5.5 ± 0.7		5.1 ± 1.0	
		Catechin	5.2 ± 1.0	0.383	5.3 ± 0.9	0.240	5.2 ± 0.9	0.891
	Reaction time (ms)	Placebo	1172 ± 291		1186 ± 190		1205 ± 195	
		Catechin	1131 ± 245	0.188	1109 ± 200	0.167	1133 ± 229	0.197

POET, Perception of Emotions Test. Values are presented as mean ± SD.

**Table 5 molecules-25-04265-t005:** Effects of green tea catechins on working memory-related tasks.

Task			−1 Week (Baseline)	*p* Values	0 Week (Single Dose)	*p* Values	12 Weeks	*p* Values	Change from Baseline (%)	
									0 Week	12 Weeks
FPCPT (Part 3)	Correct responses	Placebo	14.6 ± 2.3		15.4 ± 1.4		14.9 ± 3.1			
		Catechin	15.1 ± 1.5	0.804	15.5 ± 0.9	0.930	15.4 ± 1.5	0.707		
	Average correct response time (ms)	Placebo	557 ± 122		570 ± 90		600 ± 128			
		Catechin	513 ± 98	0.224	538 ± 92	0.270	534 ± 83	0.052		
	Incorrect responses	Placebo	0.12 ± 0.33		0.08 ± 0.28		0.60 ± 2.1			
		Catechin	0.04 ± 0.20	0.312	0.00 ± 0.00	0.180	0.09 ± 0.29	0.451		
	Average incorrect response time (ms)	Placebo	94.0 ± 280		50.5 ± 176		113 ± 284			
		Catechin	29.3 ± 149	0.322	0.0 ± 0.0	0.180	59.0 ± 210	0.473		
	Omission errors	Placebo	1.42 ± 2.3		0.64 ± 1.4		1.08 ± 3.1			
		Catechin	0.92 ± 1.5	0.804	0.52 ± 0.95	0.930	0.64 ± 1.5	0.707		
FPCPT (Part 4)	Correct responses	Placebo	12.0 ± 3.3		11.8 ± 3.3		11.6 ± 3.6			
		Catechin	12.7 ± 3.1	0.454	12.9 ± 3.4	0.121	13.4 ± 3.0	0.030		
	Average correct response time (ms)	Placebo	675 ± 108		687 ± 123		739 ± 180			
		Catechin	656 ± 169	0.328	629 ± 135	0.161	614 ± 109 *	0.012	−37.2 ± 148 (*p* = 0.322)	−52.0 ± 116 ** (*p* = 0.001)
	Incorrect responses	Placebo	1.8 ± 1.8		1.0 ± 1.3		1.6 ± 1.5			
		Catechin	1.2 ± 1.2	0.220	1.0 ± 1.0	0.671	1.2 ± 1.1	0.398		
	Average incorrect response time (ms)	Placebo	684 ± 458		514 ± 547		720 ± 487			
		Catechin	525 ± 462	0.358	541 ± 452	0.949	585 ± 411	0.250		
	Omission errors	Placebo	4.0 ± 3.3		4.2 ± 3.3		4.4 ± 3.6			
		Catechin	3.3 ± 3.1	0.454	3.1 ± 3.4	0.121	2.6 ± 3.0	0.030		

FPCPT, 4-part Continuous Performance Test. Values are presented as mean ± SD. *p* Values are calculated using unpaired *t*-tests or Mann–Whitney *U* tests and Bonferroni correction; * *p* < 0.05/3 = 0.017, ** *p* < 0.01/3 = 0.003 vs placebo group.

**Table 6 molecules-25-04265-t006:** Effects of green tea catechins on visual information processing-related tasks.

Task			−1 Week (Baseline)	*p* Values	0 Week (Single Dose)	*p* Values	12 Weeks	*p* Values
SDC	Correct responses	Placebo	55.7 ± 8.8		59.6 ± 9.2		61.5 ± 8.2	
		Catechin	57.0 ± 11	0.642	62.2 ± 9.7	0.333	61.9 ± 11	0.879
	Errors	Placebo	0.50 ± 1.1		1.04 ± 1.6		0.72 ± 0.98	
		Catechin	0.77 ± 1.2	0.249	0.70 ± 0.76	0.982	0.73 ± 1.1	0.800
NVRT	Correct responses	Placebo	9.6 ± 2.2		9.6 ± 1.9		10.1 ± 2.0	
		Catechin	9.5 ± 2.4	0.905	10.2 ± 1.9	0.289	10.0 ± 1.7	0.885
	Average correct reaction time (ms)	Placebo	4384 ± 862		4131 ± 885		4123 ± 1077	
		Catechin	4047 ± 865	0.166	4102 ± 694	0.901	3874 ± 827	0.384
	Commission errors	Placebo	5.0 ± 2.4		4.9 ± 1.9		4.5 ± 2.2	
		Catechin	5.3 ± 2.5	0.736	4.6 ± 2.1	0.586	4.7 ± 1.9	0.681
	Omission errors	Placebo	0.35 ± 0.56		0.44 ± 0.82		0.44 ± 0.71	
		Catechin	0.19 ± 0.40	0.319	0.17 ± 0.39	0.311	0.27 ± 0.46	0.486

SDC, Symbol Digit Coding Test; NVRT, Non-verbal Reasoning Test. Values are presented as mean ± SD.

**Table 7 molecules-25-04265-t007:** Effects of green tea catechins on a motor function-related task.

Task			−1 Week (Baseline)	*p* Values	0 Week (Single Dose)	*p* Values	12 Weeks	*p* Values
FTT	Right taps average	Placebo	58.7 ± 6.5		59.7 ± 5.9		57.7 ± 6.8	
		Catechin	59.5 ± 7.9	0.689	58.9 ± 7.9	0.689	57.7 ± 8.0	0.983
	Left taps average	Placebo	55.8 ± 7.1		55.4 ± 6.6		54.6 ± 7.9	
		Catechin	55.2 ± 8.2	0.787	54.8 ± 8.1	0.788	53.9 ± 7.8	0.762

FTT, Finger-tapping Test. Values are presented as mean ± SD.

**Table 8 molecules-25-04265-t008:** Effects of green tea catechins on dementia-related blood biomarkers.

		*n*	−1 Week (Baseline)	*p* Values	12 Weeks	*p* Values	Change from Baseline	*p* Values
Plasma Aβ (1–40) (pg/mL)	Placebo	24	201 ± 43		184 ± 32		−17.8 ± 32	
	Catechin	21	210 ± 51	0.446	195 ± 46	0.439	−15.0 ± 38	0.982
Plasma Aβ (1–42) (pg/mL)	Placebo	10	10.5 ± 3.3		9.94 ± 4.2		−0.52 ± 4.8	
	Catechin	13	14.6 ± 7.5	0.148	15.0 ± 9.0	0.088	0.46 ± 5.3	0.92
Aβ (1-42)/Aβ (1–40)	Placebo	10	0.054 ± 0.03		0.053 ± 0.02		−0.002 ± 0.02	
	Catechin	12	0.071 ± 0.04	0.418	0.075 ± 0.04	0.254	0.004 ± 0.03	0.923
Plasma sAPPα (ng/mL)	Placebo	24	8.5 ± 2.5		10.4 ± 2.7		1.89 ± 3.2	
	Catechin	22	9.6 ± 3.8	0.441	11.0 ± 2.7	0.660	1.41 ± 2.7	0.733
Plasma APP770 (ng/mL)	Placebo	23	30.9 ± 9.9		27.4 ± 8.7		−3.51 ± 10	
	Catechin	21	35.0 ± 12	0.200	29.8 ± 8.2	0.269	−5.12 ± 10	0.372
Serum BDNF (ng/mL)	Placebo	23	20.0 ± 17		20.9 ± 13		0.90 ± 21	
	Catechin	20	17.7 ± 10	0.846	25.7 ± 11	0.158	7.97 ± 13	0.165

Aβ, amyloid β; sAPPα, secreted form of amyloid precursor protein α; APP, amyloid precursor protein; BDNF, brain-derived neurotrophic factor. Values are presented as mean ± SD.

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
