# Peer review of "Effect of Daily Intake of Green Tea Catechins on Cognitive Function in Middle-Aged and Older Subjects: A Randomized, Placebo-Controlled Study"

_molecules, 2020, doi:10.3390/molecules25184265_

Round 1

Reviewer 1 Report

  1. The main weakness is its observational nature without any mechanistic insight of potential protective effects.
  2. The sample size is small in this study. With the small size, p < 0.01 is not statistically meaningful. (Table 5)
  3. The only significant difference between the placebo group and the Catechin was “FPCPT, 4-part Continuous Performance Test” ---- “intake of 336.4 mg of GTC maintained or improved working memory in subjects”, this conclusion seems to be overstated. There was no dose response study designed.
  4. “Catechins, which are typical polyphenols contained in green tea, have been reported to have antioxidative, anti-inflammatory, and neuroprotective effects.” However, in the study design, “The subjects were not required to restrict their intake of other polyphenols (green tea, black tea, oolong tea, etc.) and or to deviate from their usual diets.” If intake of green tea is not controlled, it is hard to compare the real level of catechins.

Reviewer 2 Report

The authors presented a randomized controlled trial investigating the effect of tea catechins administration on cognitive function. The methodology does not have any particular flaws besides the following. I would suggest few minor revisions:

- The authors should better describe how the administered dose was decided (comparison with previously published studies?) and how it compares with other observational studies on dietary polyphenols from foods (to be implemented in the discussion).

- It is important to stress the concept that the results obtained are applicable for Japanese population and further studies in caucasian individuals are needed to generalize its applicability worldwide.

- Patients were not asked to restrict their usual polyphenol intake. However, it would be important to test their dietary intake and check whether there were differences between intervention and control group. If the authors don't have this information, they should state in the limitation that this could be a source of bias.

- The authors may consider better describing in the discussion also the gut-brain axis to explain the potential mechanisms of action of tea catechins (a review for reference (PMID: 29671359)

Round 2

Reviewer 1 Report

The authors answered most of my questions and the revised manuscript was improved.